# Thermodynamic, Non-Extensive, or Turbulent Quasi-Equilibrium for the Space Plasma Environment

**DOI:** 10.3390/e21090820

**Published:** 2019-08-22

**Authors:** Peter H. Yoon

**Affiliations:** 1Institute for Physical Science and Technology, University of Maryland, College Park, MD 20742, USA; yoonp@umd.edu; 2Korea Astronomy and Space Science Institute, Daejeon 34055, Korea

**Keywords:** Kappa distribution, non-extensive entropy, plasma turbulence, solar wind, kinetic theory

## Abstract

The Boltzmann–Gibbs (BG) entropy has been used in a wide variety of problems for more than a century. It is well known that BG entropy is additive and extensive, but for certain systems such as those dictated by long-range interactions, it is speculated that the entropy must be non-additive and non-extensive. Tsallis entropy possesses these characteristics, and is parameterized by a variable *q* (q=1 being the classic BG limit), but unless *q* is determined from microscopic dynamics, the model remains a phenomenological tool. To this day, very few examples have emerged in which *q* can be computed from first principles. This paper shows that the space plasma environment, which is governed by long-range collective electromagnetic interaction, represents a perfect example for which the *q* parameter can be computed from microphysics. By taking the electron velocity distribution function measured in the heliospheric environment into account, and considering them to be in a quasi-equilibrium state with electrostatic turbulence known as quasi-thermal noise, it is shown that the value corresponding to q=9/13=0.6923, or alternatively q=5/9=0.5556, may be deduced. This prediction is verified against observations made by spacecraft, and it is shown to be in excellent agreement. This paper constitutes an overview of recent developments regarding the non-equilibrium statistical mechanical approach to understanding the non-extensive nature of space plasma, although some recent new developments are also discussed.

## 1. Introduction

The celebrated Boltzmann–Gibbs (BG) definition for entropy [1,2],
(1)$$S_{BG}=k\log W,$$
has been used in a wide variety of problems for more than a century. Here, *W* represents the combinatorial number of all possible microstates of a system, be it classical or quantum mechanical. The constant *k* is taken as the Boltzmann constant $k_{B}=1.3806503\times 10^{-23}\;m^{2}\;\mathrm{kg}\;s^{-2}\;K^{-1}$ for thermostatistics and unity for the information system, in which case it is known as the Shannon entropy [3]. A more general form of Boltzmann–Gibbs–Shannon (BGS) entropy is expressed in terms of the probability, $p_{i}$, of the system being in a particular microstate labeled *i*, namely,

(2)$$S_{BG}=-k\sum_{i=1}^{W}p_{i}\;\ln p_{i}.$$

For the particular case of equal probability, $p_{i}=1/W$ for all *i*, we recover $S_{BG}=k\log W$. This well-known expression for the entropy has been in use since the 1870s, not only in physics, but for a variety of problems in chemistry, mathematics, computational sciences, engineering, and elsewhere.

It should be noted that the BG entropy is not universally applicable to all situations. While the definition is eminently suitable for an ideal gas and systems dictated by short-range interactions, for systems interacting through long-range forces, the applicability of BG entropy has been doubted by a number of individuals, including Boltzmann himself [4], Einstein [5], Fermi [6], and others. One of the characteristics of BG entropy is that it is additive and extensive. If *A* and *B* represent two subsystems and A+B the total system, then the Boltzmann–Gibbs entropy of each system is given by $S_{BG}\left(A\right)=k\ln W_{A}$ and $S_{BG}\left(B\right)=k\ln W_{B}$, where $W_{A}$ and $W_{B}$ represent the number of possible states for systems *A* and *B*, respectively. Then it is straightforward to show that the entropy of the total system is the sum of the entropies of both systems,

(3)$$S_{BG}\left(A+B\right)=k\ln\left(W_{A}W_{B}\right)=k\ln W_{A}+k\ln W_{B}=S_{BG}\left(A\right)+S_{BG}\left(B\right).$$

This is the additive property of BG entropy. The Boltzmann entropy is also extensive. The extensivity means that the entropy is proportional to the total number of particles. For systems dictated by additive properties of entropy, it is reasonable to assume that the number of possible states behaves as
(4)$$W\left(N\right)\propto w^{N}\;\left(w>1\right),$$
where *N* represents the total number of particles. This implies that the equal probability is given by $p\left(N\right)=1/w^{N}$, from which it is straightforward to see that

(5)$$S_{BG}=Nk\ln w\propto N.$$

Hence, Boltzmann entropy is extensive. Non-additive/non-extensive entropy violates these additive and extensive properties.

For short-range interactions, microstates are governed by neighboring particles. As such, the combinatorial number of possible microstates associated with the total system is simply the sum of combinatorial number of microstates associated with each subsystem. This is because of the fact that particles interacting with short-range force are not aware of the presence of other particles belonging to other subsystems. The ionized gas, or plasma, is governed by long-range electromagnetic interaction among charged particles. As such, the combinatorial number of microstates may be more than that of the simple sum for subsystems since charged particles in one system are affected by distant particles in other subsystem by virtue of the long-range nature of the interaction. Likewise, if the total system is accessible to a combinatorial number of microstates that is greater (or less) than those of individual systems, then the total number of possible states may not simply be represented by a simple scaling behavior, $W\propto w^{N}$. For such a situation, the non-additive/non-extensive entropic principle, S(A+B)≠S(A)+S(B) or S(N)≠O(N), is expected.

Over the past many years, a number of attempts were made to generalize BG entropy to non-additive/non-extensive situations. Among these is the 1988 paper by Tsallis [7], which has triggered a recent interest in non-extensive thermostatics, although in a strict sense, Tsallis was not the first to suggest the particular functional form. As he acknowledges in his recent monograph [8], a number of previous attempts had already entertained similar functional forms for generalized entropy (see entry 107 in the bibliography of [8], p. 347, for the full references of early works.). Tsallis put forth a model entropy of the form

(6)$$S_{q}=-\frac{k\left(1-\sum_{i=1}^{W}p_{i}^{q}\right)}{1-q}.$$

It can be shown that
(7)$$\frac{S_{q}\left(A+B\right)}{k}=\frac{S_{q}\left(A\right)}{k}+\frac{S_{q}\left(B\right)}{k}+\left(1-q\right)\;\frac{S_{q}\left(A\right)}{k}\frac{S_{q}\left(B\right)}{k}\neq \frac{S(A)}{k}+\frac{S(B)}{k},$$
where the parameter *q* is a measure of how far the system deviates from the BG statistics (for which q=1). For equal probability, $W\left(N\right)=w^{N}$ and $p\left(N\right)=w^{-N}$, Tsallis entropy is given by
(8)$$\frac{S_{q}\left(N\right)}{k}=\frac{1}{q-1}\left[1-w^{-N(q-1)}\right],$$
which is not proportional to *N*, and hence, is non-extensive. Note that a number of modifications and improvements of Tsallis’ original formalism are available in the literature [9], but the original formalism by Tsallis is sufficient for the present discussion.

An outstanding issue concerns the determination of the *q* parameter from first principles. Chapter 7 of the monograph by Tsallis [8] lists applications of the non-extensive statistical method to high-energy physics, turbulence, granular matter, geophysics and astrophysics, quantum chaos, etc., where the *q* parameter for each case is determined by empirical fitting method. For plasmas, on the other hand, the governing principle (that is, the laws of electromagnetics) is simple enough that it is possible to compute the value of the *q* parameter from first principles. The space environment is an almost perfect natural laboratory for plasma research, so we focus on examples from measurements made in space by artificial satellites in order to investigate the basic property of space plasmas and a possible connection to non-extensive statistical concepts.

In the 1960s, in situ spacecraft measurements of charged particles became possible. It was realized then that the space plasma did not behave according to the laws of thermodynamic equilibrium. Instead of a Maxwell–Boltzmann–Gauss distribution of particles, observed distributions typically featured a suprathermal component, see, e.g., [10,11,12] for some early observations. In an attempt to fit the measured electron distributions, Vasyliunas [13] introduced a phenomenological model known as the kappa distribution,
(9)$$f_{\kappa }^{\mathrm{Vasyliunas}}\left(v\right)∼{\left(1+\frac{v^{2}}{\kappa v_{T}^{2}}\right)}^{-(\kappa +1)},$$
where $v_{T}={\left(2k_{B}T/m\right)}^{1/2}$ is the Maxwellian thermal speed, that is, $v_{T}$ would be the thermal speed had f(v) been given by the Maxwell–Boltzmann distribution, *T* and *m* are the temperature and mass of the charged particles, κ is a free fitting parameter, and f(v) represents the velocity distribution function. When κ→∞, the model reduces to the Maxwellian–Boltzmann (or thermal) distribution,

(10)$$f_{MB}\left(v\right)∼\exp\left(-\frac{v^{2}}{v_{T}^{2}}\right).$$

The kappa model enjoyed no first principle justification until it was realized that the most probable state defined in the context of the non-extensive entropic principle is related to the kappa distribution. For a continuous system, BG entropy can be written as
(11)$$S_{BG}=-k_{B}\int dx\int dv\;f\left(v\right)\;\ln f\left(v\right),$$
where integration over space ∫dx is understood as being normalized with respect to the total volume of the system, $V^{-1}\int dx$. Upon minimizing the Helmholtz free energy,
(12)$$F=U-TS_{BG},$$
where
(13)$$U=\int dx\int dv\;\frac{mv^{2}}{2}\;f\left(v\right)$$
is the total energy, we find that the Maxwell–Boltzmann distribution naturally emerges as the most probable state. In contrast, upon making use of the continuous version of the non-extensive (NE), or Tsallis entropy, to wit,
(14)$$S_{q}=-\frac{k_{B}}{1-q}\int dx\int dv\left\{f\left(v\right)-{\left[f\left(v\right)\right]}^{q}\right\},$$
and minimizing the corresponding Helmholtz free energy, the solution
(15)$$f_{q}\left(v\right)∼{\left[1+\frac{\left(1-q\right)\;v^{2}}{v_{T}^{2}}\right]}^{-1/(1-q)}\equiv \exp_{q}\left(\frac{v}{v_{T}}\right)$$
emerges as the most probable state. Here, $\exp_{q}\left(x\right)={\left[1+\left(1-q\right)x^{2}\right]}^{-1/(1-q)}$ is known as the *q*-exponential function. Upon defining
(16)$$\kappa =\frac{1}{1-q}\;\mathrm{or}\;\kappa =\frac{q}{1-q},$$
it is straightforward to see that this solution is either $f∼{\left[1+v^{2}/\left(\kappa v_{T}^{2}\right)\right]}^{-\kappa }$ or $f∼{\left\{1+v^{2}/\left[\left(\kappa +1\right)v_{T}^{2}\right]\right\}}^{-\kappa +1}$, either of which is similar to Vasyliunas’ kappa distribution. Note, however, that Vasyliunas’ original model has the asymptotic power-law index κ+1 and the parameter κ associated with thermal speed, $f∼{\left[1+v^{2}/\left(\kappa v_{T}^{2}\right)\right]}^{-\kappa +1}$, so that neither choice strictly reproduces Vasyliunas’ original model. Nonetheless, the two alternative forms for *f* and Vasyliunas’ model are practically equivalent. This has prompted an explosion of interest in the non-extensive statistical model in the framework of space plasmas [14,15,16].

Independent of these developments, ref. [17] uncovered an interesting fact that the quasi steady state electrostatic turbulence generated by a weak electron beam propagating in the background plasma is characterized by a non-thermal tail population in the electron velocity distribution function, which superficially resembles the kappa distribution. By “quasi-equilibrium”, we designate a steady-state turbulence but one that has not yet reached a genuine thermodynamic equilibrium via collisional relaxation. Subsequently, ref. [18] provided further proof that the kappa distribution is a unique solution that characterizes steady-state electrostatic plasma turbulence. The finding that quasi-steady-state plasma turbulence corresponds to the kappa distribution function strongly implies the profound inter-relationship with a non-extensive statistical equilibrium. Quasi-equilibrium states of plasma turbulence depict electrons interacting amongst themselves through long-range collective electrostatic fluctuations, which also describes the non-extensive charged-particle system interacting with long-range electrostatic force. In this regard, it is no surprise that the approach based upon plasma turbulence theory and that based on non-extensive statistical method are equivalent.

Solar wind electrons [19,20] can be modeled with multi-component velocity distribution functions: Maxwellian core; suprathermal halo; highly field-aligned strahl, which is typically associated with high-speed wind; and highly energetic superhalo. As the superhalo electrons [21], which are observed in all solar wind conditions with a nearly invariant velocity power-law index, are at the high end of the velocity spectrum, comparing the asymptotic behavior of the kappa distribution with that of superhalo component can yield meaningful results, which we will do later in this paper. For the moment, we move on to the next section, where we discuss the asymptotically steady-state electrostatic plasma turbulence and the corresponding electron velocity distribution.

It is the purpose of the present paper to overview recent developments in the theory of the electron kappa distribution in space from the perspective of non-equilibrium statistical mechanics. As such, the next few sections largely recapitulate and summarize known results, especially those of ref. [18], although the presentation of plasma kinetic-theory-based results in conjunction with non-extensive thermodynamic concepts, as we undertake in this paper, has not been done before. Some new developments will also be discussed, including the question of the transition from the kappa distribution to an eventual thermal distribution and the modification of the standard kappa distribution as well as its consequences on observed charged particle distributions in space.

## 2. Steady-State Plasma Turbulence and the Electron Kappa Distribution

We begin the discussion with the kinetic equations of plasma turbulence, which are available in the standard plasma physics literature [17,22,23,24,25]. The basic derivation begins with the standard microscopic kinetic equation for plasma dynamics, namely, the exact phase-space mapping equation known as the Klimontovich equation, which describes the time evolution of a 6N dimensional phase-space distribution function,

(17)$$N\left(r,v,t\right)=\sum_{i=1}^{N}\delta \left[r-r_{i}\left(t\right)\right]\delta \left[v-v_{i}\left(t\right)\right].$$

The exact particle orbit $r_{i}\left(t\right)$ and $v_{i}\left(t\right)$ satisfies the classical Lorentz equation,
(18)$$\begin{array}{ccc} \dot{r}\left(t\right) & = & v_{i}\left(t\right),\\ \dot{v}\left(t\right) & = & \frac{F[r_{i}\left(t\right),v_{i}\left(t\right)]}{m}, \end{array}$$
where $F\left(r,v\right)=e_{i}E+\left(e/c\right)v\times B$ is the Lorentz force, E and B being the electric and magnetic field vectors, respectively. Of course, E and B satisfy Maxwell’s equation, where the total charge and current are determined by

(19)$$\begin{array}{ccc} \rho & = & e\int dv\;N(r,v,t),\\ j & = & e\int dv\;vN(r,v,t). \end{array}$$

Upon separating physical quantities into averaged and fluctuating quantities, N(r,v,t)=nf(r,v,t)+δN(r,v,t), $E=E_{0}+\delta E$, $B=B_{0}+\delta B$, and carrying out the perturbative analysis under the assumption of small amplitude fluctuations, that is, weak turbulence, and taking ensemble averages (or averages over random phases of the fluctuation), it can be shown that the exact phase-space Klimontovich kinetic equation,
(20)$$\frac{dN}{dt}=\left(\frac{\partial }{\partial t}+v\cdot \nabla +\frac{F}{m}\cdot \frac{\partial }{\partial v}\right)N=0,$$
which is microscopically time reversible, reduces to the equations of non-equilibrium statistical mechanics that represent time-irreversible processes. For plasmas in a uniform medium free of external fields $\left(E_{0}=0=B_{0}\right)$ and subject to electrostatic interactions among charged particles (that is, δB=0) including dynamic electrons and quasi-stationary neutralizing background protons, it can be shown that the electron distribution function $f_{e}\left(v,t\right)$ obeys the kinetic equation given by

(21)$$\begin{array}{ccc} \frac{\partial f_{e}}{\partial t} & = & \frac{\partial }{\partial v_{i}}\left(A_{i}F_{e}+D_{ij}\;\frac{\partial f_{e}}{\partial v_{j}}\right),\\ A_{i} & = & \frac{e^{2}}{4\pi m_{e}}\int dk\;\frac{k_{i}}{k^{2}}\sum_{\sigma =\pm 1}\sigma \omega_{k}^{L}\;\delta \left(\sigma \omega_{k}^{L}-k\cdot v\right),\\ D_{ij} & = & \frac{\pi e^{2}}{m_{e}^{2}}\int dk\;\frac{k_{i}\;k_{j}}{k^{2}}\sum_{\sigma =\pm 1}\delta \left(\sigma \omega_{k}^{L}-k\cdot v\right)\;I_{k}^{\sigma L}. \end{array}$$

In the above, *e* and $m_{e}$ stand for the unit electric charge and electron mass, respectively,
(22)$$\omega_{k}^{L}=\omega_{pe}\left(1+\frac{3}{2}\;k^{2}\lambda_{De}^{2}\right),$$
represents the dispersion relation satisfied by a high-frequency electrostatic wave in the plasma known as the Langmuir wave, $\lambda_{De}=T_{e}^{1/2}/{\left(4\pi ne^{2}\right)}^{1/2}$ is the Debye length, $\omega_{pe}={\left(4\pi ne^{2}/m_{e}\right)}^{1/2}$ is the plasma oscillation frequency, $T_{e}$ is the electron temperature, and $I_{k}^{\sigma L}$ denotes the spectral electric field intensity associated with the Langmuir wave, $E_{k,\omega }^{2}=\sum_{\sigma =\pm 1}I_{k}^{\sigma L}\delta \left(\omega -\sigma \omega_{k}^{L}\right)$. The symbol σ=±1 denotes the sign of the wave phase and group velocities. The electron particle kinetic Equation (Equation 21) is given in the form of the velocity-space Fokker–Planck equation, where the velocity friction represented by coefficient A appears in balanced form against velocity diffusion with the associated diffusion coefficient $D_{ij}$. Note that the above form of the Fokker–Planck equation is governed by wave–particle interactions rather than collisional processes. It is well known that if the plasma is governed by binary collisions among charged particles, then the resulting collisional kinetic equation, or generalized Boltzmann equation (in plasmas it is known as the Landau collisional equation or more generally as the Balescu–Lenard collisional equation), is also given by the Fokker–Planck form. It contrast, the Fokker–Planck Equation (Equation 21) is dictated by wave–particle interactions, as evidenced by the resonance condition $\sigma \omega_{k}^{L}-k\cdot v=0$ within the definitions for $A_{i}$ and $D_{ij}$.

Since the velocity diffusion is dictated by the Langmuir wave spectral intensity $I_{k}^{\sigma L}$, the particle equation must be considered in conjunction with the wave kinetic equation. When one considers the wave kinetic equation, one must take into account not only the high-frequency Langmuir (*L*) wave but also the low-frequency wave known as the ion-sound (*S*) wave, since the *L* mode is nonlinearly coupled to the *S* mode via a wave–wave resonant interaction. In short, the wave kinetic equations for the *L* and *S* mode waves are to be solved as well. Following the procedures outlined at the beginning of this section, it can be shown that the wave kinetic equation for Langmuir and ion-sound wave intensities are given by [17,22,23,24,25]

(23)$$\begin{array}{lll} \frac{\partial I_{k}^{\sigma L}}{\partial t} & = & \frac{\pi \omega_{pe}^{2}}{k^{2}}\int dv\;\delta \left(\sigma \omega_{k}^{L}-k\cdot v\right)\left(\frac{ne^{2}}{\pi }\;f_{e}+\sigma \omega_{k}^{L}\;I_{k}^{\sigma L}\;k\cdot \frac{\partial f_{e}}{\partial v}\right)\\ & & +2\sum_{\sigma^{\prime },\sigma^{\prime \prime }=\pm 1}\sigma \omega_{k}^{L}\int dk^{\prime }\;V_{k,k^{\prime }}^{LS}\left(\sigma \omega_{k}^{L}\;I_{k^{\prime }}^{\sigma^{\prime }L}\;I_{k-k^{\prime }}^{\sigma^{\prime \prime }S}\right.\\ & & \left.-\sigma^{\prime }\omega_{k^{\prime }}^{L}\;I_{k-k^{\prime }}^{\sigma^{\prime \prime }S}\;I_{k}^{\sigma L}-\sigma^{\prime \prime }\omega_{k-k^{\prime }}^{L}\;I_{k^{\prime }}^{\sigma^{\prime }L}\;I_{k}^{\sigma L}\right)\\ & & +\sigma \omega_{k}^{L}\sum_{\sigma^{\prime }=\pm 1}\int dk^{\prime }\int dv\;U_{k,k^{\prime }}^{LL}\left[\frac{m_{e}}{m_{i}}\;I_{k^{\prime }}^{\sigma^{\prime }L}\;I_{k}^{\sigma L}\;\left(k-k^{\prime }\right)\cdot \frac{\partial f_{i}}{\partial v}\right.\\ & & \left.+\frac{ne^{2}}{\pi \omega_{pe}^{2}}\left(\sigma \omega_{k}^{L}\;I_{k^{\prime }}^{\sigma^{\prime }L}-\sigma^{\prime }\omega_{k^{\prime }}^{L}\;I_{k}^{\sigma L}\right)\left(f_{e}+f_{i}\right)\right], \end{array}$$

(24)$$\begin{array}{lll} \frac{\partial I_{k}^{\sigma S}}{\partial t} & = & \frac{\pi \mu_{k}\omega_{pe}^{2}}{k^{2}}\int dv\;\delta \left(\sigma \omega_{k}^{S}-k\cdot v\right)\left[\frac{ne^{2}}{\pi }\;\left(f_{e}+f_{i}\right)\right.\\ & & \left.+\;\sigma \omega_{k}^{L}\;I_{k}^{\sigma S}\left(k\cdot \frac{\partial }{\partial v}\right)\left(f_{e}+\frac{m_{e}}{m_{i}}\;f_{i}\right)\right]\\ & & +\sum_{\sigma^{\prime },\sigma^{\prime \prime }=\pm 1}\sigma \omega_{k}^{L}\;\int dk^{\prime }\;V_{k,k^{\prime }}^{SL}\left(\sigma \omega_{k}^{L}\;I_{k^{\prime }}^{\sigma^{\prime }L}\;I_{k-k^{\prime }}^{\sigma^{\prime \prime }L}\right.\\ & & \left.-\sigma^{\prime }\omega_{k^{\prime }}^{L}\;I_{k-k^{\prime }}^{\sigma^{\prime \prime }L}\;I_{k}^{\sigma S}-\sigma^{\prime \prime }\omega_{k-k^{\prime }}^{L}\;I_{k^{\prime }}^{\sigma^{\prime }L}\;I_{k}^{\sigma S}\right). \end{array}$$

For the *L* mode wave equation, the first term on the right-hand side represents the linear wave–particle resonant interaction between the electrons and the Langmuir wave; the second and third lines collectively describe three wave or wave–wave nonlinear resonant processes among two Langmuir waves and an ion sound wave; the fourth and fifth lines together describe nonlinear wave–particle resonance among two Langmuir waves mediated by quasi-stationary protons whose velocity distribution is given by $f_{i}$. For the ion sound mode, the interpretations and designations of various terms are analogous to those of the *L* mode wave, except that the *S* mode wave kinetic equation does not have the term denoting the nonlinear wave–particle resonance. The ion sound or *S* mode enjoys the dispersion relation specified by
(25)$$\omega_{k}^{S}=\frac{kc_{S}\sqrt{1+3T_{i}/T_{e}}}{\sqrt{1+k^{2}\lambda_{De}^{2}}},$$
where $c_{S}={\left(T_{e}/m_{i}\right)}^{1/2}$ is the ion sound (or ion acoustic) speed and $T_{i}$ stands for the ion (proton) temperature. Various objects that appear in the wave kinetic Equations (Equation 23) and (Equation 24) are defined by

(26)$$\begin{array}{ccc} \mu_{k} & = & k^{3}\lambda_{De}^{3}\sqrt{\frac{m_{e}}{m_{i}}}\sqrt{1+\frac{3T_{i}}{T_{e}}},\\ V_{k,k^{\prime }}^{LS} & = & \frac{\pi }{2}\frac{e^{2}}{T_{e}^{2}}\frac{\mu_{k-k^{\prime }}\;{\left(k\cdot k^{\prime }\right)}^{2}}{k^{2}\;k^{\prime 2}{\left|k-k^{\prime }\right|}^{2}}\;\delta \left(\sigma \omega_{k}^{L}-\sigma^{\prime }\omega_{k^{\prime }}^{L}-\sigma^{\prime \prime }\omega_{k-k^{\prime }}^{S}\right),\\ V_{k,k^{\prime }}^{SL} & = & \frac{\pi }{4}\frac{e^{2}}{T_{e}^{2}}\frac{\mu_{k}\;{\left[k^{\prime }\cdot \left(k-k^{\prime }\right)\right]}^{2}}{k^{2}\;k^{\prime 2}\;{\left|k-k^{\prime }\right|}^{2}}\;\delta \left(\sigma \omega_{k}^{S}-\sigma^{\prime }\omega_{k^{\prime }}^{L}-\sigma^{\prime \prime }\omega_{k-k^{\prime }}^{L}\right),\\ U_{k,k^{\prime }}^{LL} & = & \frac{\pi }{\omega_{pe}^{2}}\frac{e^{2}}{m_{e}^{2}}\frac{{\left(k\cdot k^{\prime }\right)}^{2}}{k^{2}\;k^{\prime 2}}\;\delta \left[\sigma \omega_{k}^{L}-\sigma^{\prime }\omega_{k^{\prime }}^{L}-\left(k-k^{\prime }\right)\cdot v\right]. \end{array}$$

As is apparent from the definitions, nonlinear coupling coefficients, $V_{k,k^{\prime }}^{LS}$, $V_{k,k^{\prime }}^{SL}$, and $U_{k,k^{\prime }}^{LL}$, dictate the various wave–wave and nonlinear wave–particle resonant interactions, which are obvious from the delta function arguments.

Consider the particle kinetic equation for electrons (Equation 21). In what follows, we assume that the wave dispersion relation depends only on the magnitude of k and that the forward and backward wave intensities are identical and isotropic, $\omega_{k}^{L}=\omega_{k}^{L}$ and $I_{k}^{\sigma L}=I_{L}\left(k\right)$, which are valid assumptions provided the electron distribution function is isotropic, $f_{e}\left(v\right)=f_{e}\left(v\right)$. We assume the steady state, $\partial f_{e}/\partial t=0$, by virtue of the velocity friction and diffusion terms balancing each other out,

(27)$$0=A_{i}f_{e}+D_{ij}\;\frac{\partial f_{e}}{\partial v_{j}}.$$

Following the basic method pioneered in ref. [26], the present author [18] demonstrated that the formal solution to the steady-state particle kinetic equation is given by

(28)$$f_{e}=C\;\exp\left(-\int dv\;\frac{m_{e}v}{4\pi^{2}}\frac{\int_{\omega_{pe}/v}^{\infty }\frac{dk}{k}}{\int_{\omega_{pe}/v}^{\infty }\frac{dk}{k}\;I_{L}\left(k\right)}\right).$$

In the above, the integral $\int_{\omega_{pe}/v}^{\infty }dk/k$ formally diverges for k→∞, but if we formally define

(29)$$H\left(v\right)=\int_{\omega_{pe}/v}^{\infty }\frac{dk}{k},\;H\left(v\right)\;I\left(v\right)=\int_{\omega_{pe}/v}^{\infty }\frac{dk}{k}\;I_{L}\left(k\right).$$

Then we may formally remove the divergence, so that we have

(30)$$f_{e}=C\;\exp\left(-\int dv\;\frac{m_{e}v}{4\pi^{2}}\frac{1}{I(v)}\right).$$

This solution shows that a suitable model for $I_{L}\left(k\right)$, or for that matter, I(k), will lead to the suitable counterpart for $f_{e}$ and vice versa. Apparently, there exists an infinite choice for coupled solutions $\left[f_{e}\left(v\right),I_{L}\left(k\right)\right]$, of which we are interested in a particular form of electron velocity distribution function that represents a kappa-like solution,

(31)$$f_{e}\left(v\right)=\frac{C}{{\left(1+m_{e}v^{2}/2\kappa^{\prime }\;\theta_{e}\right)}^{\kappa +1}}.$$

The normalization constant *C* can be obtained by requiring the condition $1=4\pi \int_{0}^{\infty }dv\;v^{2}f_{e}$ and is thus given by

(32)$$C=\frac{m_{e}^{3/2}}{{\left(2\pi \;\theta_{e}\right)}^{3/2}}\frac{\Gamma (\kappa +1)}{{\kappa^{\prime }}^{3/2}\;\Gamma \left(\kappa -1/2\right)}.$$

Here, Γ(x) is the gamma function. The effective or kinetic temperature for this kappa-like model can be computed on the basis of definition, $T_{e}=\int dv\left(m_{e}v^{2}/3\right)f_{e}$, and the result is

(33)$$T_{e}=\theta_{e}\;\frac{\kappa^{\prime }}{\kappa -3/2}.$$

If we impose the model $f_{e}$ given by (Equation 31), then it follows from (Equation 28) or (Equation 30) that the corresponding wave spectrum $I_{L}\left(k\right)$ can be deduced. First, it can be shown that the choice of
(34)$$I\left(v\right)=\frac{\theta_{e}}{4\pi^{2}}\frac{\kappa^{\prime }}{\kappa +1}\left(1+\frac{m_{e}v^{2}}{2\kappa^{\prime }\theta_{e}}\right),$$
satisfies (Equation 30) with $f_{e}$ given by (Equation 31). Then from (Equation 29), it follows that $I_{L}\left(k\right)$ is given by

(35)$$\begin{array}{ccc} I_{L}\left(k\right) & = & \frac{\theta_{e}}{4\pi^{2}}\frac{\kappa^{\prime }}{\kappa +1}\left(1+\frac{m_{e}\omega_{pe}^{2}}{2\kappa^{\prime }k^{2}\theta_{e}}\;\left[1+2H\left(k\right)\right]\right),\\ H(k) & = & \int_{k}^{\infty }\frac{dk}{k}. \end{array}$$

We reiterate that the distribution (Equation 31) and the corresponding spectrum (Equation 35) are not unique and that the indices κ and $\kappa^{\prime }$ are free parameters at this point. In order to prove the uniqueness as well as to determine the values for κ and $\kappa^{\prime }$, we next turn to the steady-state wave equations.

Consider the wave kinetic Equations (Equation 23) and (Equation 24). We assume an isotropic spectrum as in the above discussion on formal particle equations. We may ignore the *S* mode contribution as well as contributions from the three wave interaction process. Reference [18] discusses these assumptions and approximations in detail. The same reference also presents detailed modifications of the nonlinear coupling coefficient when the underlying electron distribution is given by the kappa-like model (Equation 31). Consequently, by omitting the intermediate steps, we simply present the steady-state Langmuir wave equation without the wave–wave resonant interaction term,
(36)$$\begin{array}{lll} 0 & = & \frac{\pi \omega_{pe}^{2}}{k^{2}}\int dv\;\delta \left(\omega_{k}-k\cdot v\right)\left(\frac{ne^{2}}{\pi }\;f_{e}+\omega_{k}\;I_{L}\left(k\right)\;k\cdot \frac{\partial f_{e}}{\partial v}\right)\\ & & -{\left(\frac{\kappa -1/2}{\kappa^{\prime }}\right)}^{2}\frac{\omega_{k}}{4\pi nT_{i}}\sum_{+,-}\int dk^{\prime }\int dv\;\frac{{\left(k\cdot k^{\prime }\right)}^{2}}{k^{2}\;k^{\prime 2}}\;\delta \left[\omega_{k}\mp \omega_{k^{\prime }}-\left(k-k^{\prime }\right)\cdot v\right]\\ & & \times \left(\frac{T_{i}}{4\pi^{2}}\left[\pm \omega_{k^{\prime }}\;I_{L}\left(k\right)-\omega_{k}\;I_{L}\left(k^{\prime }\right)\right]+I_{L}\left(k^{\prime }\right)\;I_{L}\left(k\right)\;\left(\omega_{k}\mp \omega_{k^{\prime }}\right)\right)\;f_{i}, \end{array}$$
where we have taken advantage of the fact that the ion distribution is assumed to remain stationary and is given by the Maxwellian form. Reference [18] shows that the first term on the right-hand side, which is dictated by the linear wave–particle resonance delta function condition, vanishes if $f_{e}$ and $I_{L}\left(k\right)$ are chosen by (Equation 31) and (Equation 35), respectively. As a consequence, only the nonlinear wave–particle resonance term needs to be considered in the wave equation, which provides the necessary constraint, and with which we will be able to demonstrate that the kappa-like model (Equation 31) is indeed a unique solution and that κ and $\kappa^{\prime }$ can be determined. Thus, we consider the nonlinear term in (Equation 36), which upon being set to equal to zero, reduces to

(37)$$\begin{array}{lll} 0 & = & \int dk^{\prime }\int dv\;\frac{{\left(k\cdot k^{\prime }\right)}^{2}}{k^{2}\;k^{\prime 2}}\;\delta \left[\omega_{k}-\omega_{k^{\prime }}-\left(k-k^{\prime }\right)\cdot v\right]\\ & & \times \left(\frac{T_{i}}{4\pi^{2}}\left[\omega_{k^{\prime }}\;I_{L}\left(k\right)-\omega_{k}\;I_{L}\left(k^{\prime }\right)\right]+I_{L}\left(k^{\prime }\right)\;I_{L}\left(k\right)\;\left(\omega_{k}-\omega_{k^{\prime }}\right)\right)f_{i}. \end{array}$$

As discussed in ref. [18], the nonlinear resonance speed satisfying the condition $\omega_{k}-\omega_{k^{\prime }}-\left(k-k^{\prime }\right)\cdot v=0$ is given by $v_{\mathrm{res}}∼3\left(k-k^{\prime }\right)\theta_{e}/\left(2m_{e}\omega_{pe}\right)∼v_{Ti}\ll v_{Te}$. Consequently, k and $k^{\prime }$ must be sufficiently close to each other, or $|k-k^{\prime }|∼|\delta k|\ll 1$. We thus employ the Taylor series expansion to obtain

(38)$$\begin{array}{lll} 0 & = & \int d\left(\delta k\right)\int dv\;\frac{{\left(k\cdot k^{\prime }\right)}^{2}}{k^{2}\;k^{\prime 2}}\;\delta \left[\omega_{k}-\omega_{k^{\prime }}-\left(k-k^{\prime }\right)\cdot v\right]\\ & & \times \;\left(\delta k\right)\left(\omega \left(k\right)\;\frac{dI_{L}\left(k\right)}{dk}+\frac{4\pi^{2}}{T_{i}}\;\frac{d\omega (k)}{dk}\;{\left[I_{L}\left(k\right)\right]}^{2}-\frac{d\omega (k)}{dk}\;I_{L}\left(k\right)\right)f_{i}. \end{array}$$

The necessary condition for equality leads to the spectrum
(39)$$I_{L}\left(k\right)=\frac{T_{i}}{4\pi^{2}}\left(1+\frac{2}{3}\frac{\kappa -3/2}{\kappa^{\prime }\;k^{2}\theta_{e}/m_{e}\omega_{pe}^{2}}\right),$$
which is alternative to the earlier solution (Equation 35). Obviously, the two expressions must be identical. If we identify
(40)$$\kappa =\frac{9}{4}+\frac{3}{2}\;H=2.25+1.5H,\;\kappa^{\prime }\theta_{e}=\left(\kappa +1\right)\;T_{i},$$
then we may reconcile the two expressions, where we have treated H as constant. Such a reconciliation between (Equation 35) and (Equation 39) would not have been possible had we chosen an $f_{e}$ other than the kappa-like model (Equation 31). This amounts to the uniqueness proof for the kappa distribution as being associated with the steady-state Langmuir turbulence.

To summarize the findings, the electron kappa distribution function represents a plasma state in quasi-equilibrium with weak Langmuir turbulence, and the desired final forms of $f_{e}$ and $I_{L}\left(k\right)$ are given by

(41)$$\begin{array}{ccc} f_{e}\left(v\right) & = & \frac{m_{e}^{3/2}}{{\left(2\pi T_{e}\right)}^{3/2}}\frac{\Gamma (\kappa +1)}{{\left(\kappa -3/2\right)}^{3/2}\Gamma \left(\kappa -1/2\right)}{\left(1+\frac{1}{\kappa -3/2}\frac{m_{e}v^{2}}{2T_{e}}\right)}^{-\kappa -1},\\ I_{L}\left(k\right) & = & \frac{T_{e}}{4\pi^{2}}\frac{\kappa -3/2}{\kappa +1}\left(1+\frac{1+2H}{\kappa -3/2}\frac{2\pi ne^{2}}{k^{2}T_{e}}\right),\\ \kappa & = & \frac{9}{4}+\frac{3H}{2}=2.25+1.5H,\;\frac{T_{i}}{T_{e}}=\frac{\kappa -3/2}{\kappa +1}=\frac{3+6H}{13+6H}. \end{array}$$

This solution is indirect evidence that the turbulent equilibrium in plasmas may be equivalent to the non-extensive statistical state. As we have pointed out in the Introduction, the most probable state that maximizes the Tsallis non-extensive entropy is the *q*-exponential distribution, which is equivalent to kappa distribution function. The steady state of plasma turbulence is also characterized by the same kappa distribution, which thus indicates that the two approaches are describing the same statistical state.

The kappa electron velocity distribution function may also characterize solar wind. For the suprathermal velocity range, $v\gg v_{Te}$, the kappa electron distribution (Equation 41) behaves as an inverse power-law distribution,
(42)$$f_{e}∼v^{-6.5},$$
since κ≈9/4=2.25, assuming H can be ignored. If we recall that while solar wind electrons can be modeled by a combination of Maxwellian core, suprathermal halo, and superhalo, it is the comparison with the superhalo that is most useful since these electrons are at the high end of the velocity spectrum [19,20]. Observations near Earth’s orbit show that superhalo electrons behave as $f_{e}∼v^{-5.0}$ to $v^{-8.7}$ with average behavior [21]

(43)$$f_{e}^{\mathrm{obs}}∼v^{-6.69},\;v\gg v_{Te}.$$

This agrees quite well with (Equation 42). There exists further evidence to support our interpretation that solar wind electrons are in a turbulent equilibrium state with high-frequency Langmuir fluctuations, or equivalently, they can be characterized by the non-extensive statistical state. Reference [27] analyzed solar wind halo electrons and analyzed Helios, Cluster, and Ulysses spacecraft data. The authors show that the value of the observed κ decreases from ∼9 near 0.3 AU to ∼4 near 1 AU, to ∼2.25 near ∼5 AU. This strongly implies that as solar wind evolves radially and thus approaches the quasi-equilibrium state, the distinction between halo and superhalo electrons disappears and the κ index approaches closer and closer to the theoretically predicted value.

Note that in a recent paper, Tong et al. [28] presented observational evidence that conditions favorable for whistler heat flux instability closely associate with the energized electrons featuring quasi-inverse-law velocity distribution. The whistler heat flux instability is a low frequency electromagnetic mode while Langmuir wave instability pertains to high-frequency electrostatic modes. Nevertheless, ref. [28] demonstrated the importance of wave–particle interactions in solar wind, which may lead to the locally generated energized electron distribution function. The situation observationally tested by Tong et al. [28] is highly reminiscent of the theoretical scenario envisioned by Kim et al. [29], who employed the same methodology as presented in the present paper and applied it to the whistler wave turbulence spectrum. However, it should be noted that the analysis by Kim et al. [29] is not complete since the balance of the nonlinear wave–particle interaction term is not carried out.

## 3. The Question of a True Thermodynamic Equilibrium for Space Plasma

We have thus far argued that space plasma in the heliosphere may be in a state of quasi-equilibrium in which the particles constantly exchange momentum and energy with long-range collective fluctuations, thus maintaining the kappa distribution function. Such a state, while a steady state, may not be in true thermodynamic equilibrium, which is attained through collisional relaxation. By inference to the non-extensive entropic principle, we have also conjectured that the turbulent quasi-equilibrium for space plasma may be alternatively described within the framework of the non-extensive statistical concept. The question that naturally arises is the problem of true thermodynamic equilibrium, which may eventually be reached over a much longer time period when compared with the steady-state turbulence time scale. Whether space plasma can ever attain such a state is a separate issue. For collision-poor space plasmas, the true thermodynamic equilibrium state may never be reached, even if we take into account the entire time it takes for a parcel of plasma emerging from the solar source to reach the boundary of the heliosphere, but from a theoretical point of view, a turbulent quasi-equilibrium state must eventually relax to the true thermodynamic equilibrium state through binary collisions. The road to true thermodynamic equilibrium can be discussed on the basis of kinetic plasma theory, but the analysis requires the knowledge of the time scales associated with collisional relaxation, as opposed to the time scales that govern the formation of turbulent quasi-equilibrium states. In general, it is expected that the collisional relaxation time scale is much longer than that of the turbulent equilibrium formation time scale, but the quantitative estimate is not so easy.

At the moment, we are not able to address the issue of the time scales of the collisional relaxation process. However, it is possible to discuss the theoretical framework that includes both turbulent quasi-equilibrium states and collisionally relaxed thermodynamic states within a single framework of a steady-state plasma equation. This is done by generalizing the particle kinetic equation through the addition of a collisional operator. The basic theory may be developed on the basis of the electron kinetic equation that includes the influence of collective (Langmuir wave) fluctuations and binary collisions, which can be found in ref. [30,31],
(44)$$\frac{\partial f_{e}}{\partial t}=\frac{1}{v^{2}}\frac{\partial }{\partial v}\left[v^{2}\left(A_{v}+A_{v}^{c}\right)f_{e}\right]+\frac{1}{v^{2}}\frac{\partial }{\partial v}\left(v^{2}\left(D_{vv}+D_{vv}^{c}\right)\frac{\partial f_{e}}{\partial v}\right),$$
where the particle kinetic equation that generalizes (Equation 21) for the electrons is now expressed in a spherical velocity coordinate, and we have assumed a priori that $f_{e}$ is isotropic. The velocity space friction and diffusion coefficients, A and $D_{ij}$, respectively, are the same as those defined in (Equation 21), and the additional coefficients $A^{c}$ and $D_{ij}^{c}$ pertain to collisional effects. Non-vanishing elements of these coefficients are given in spherical coordinate variables as follows:(45)$$\begin{array}{ccc} A_{v} & = & \frac{e^{2}\omega_{pe}^{2}}{m_{e}v^{2}}\int_{\omega_{pe}/v}^{\infty }\frac{dk}{k},\\ D_{vv} & = & \frac{4\pi^{2}e^{2}\omega_{pe}^{2}}{m_{e}^{2}v^{3}}\int_{\omega_{pe}/v}^{\infty }\frac{dk}{k}\;I_{L}\left(k\right),\\ A_{v}^{c} & = & \frac{4\pi ne^{4}\ln\Lambda }{m_{e}^{2}}\frac{2}{v_{Te}^{2}}\left(G\left(x_{e}\right)+\frac{T_{e}}{T_{i}}\;G\left(x_{i}\right)\right),\\ D_{vv}^{c} & = & \frac{4\pi ne^{4}\ln\Lambda }{m_{e}^{2}}\frac{G\left(x_{e}\right)+G\left(x_{i}\right)}{v},\\ x_{e} & = & \frac{v}{v_{Te}},\;x_{i}=\frac{v}{v_{Ti}},\;\Lambda =4\pi n\lambda_{De}^{3},\\ G(x) & = & \frac{\mathrm{erf}\left(x\right)-\left(2/\sqrt{\pi }\right)\;x\;e^{-x^{2}}}{2x^{2}}. \end{array}$$

In the collisional coefficients defined here, we took the approach of treating the collisional processes that involve electrons scattering off the Maxwellian distribution of charged particles via Rosenbluth potential approximation [32].

The steady-state solution is given by

(46)$$\begin{array}{lll} f_{e} & = & \mathrm{const}\;\exp\left(-\int dv\;\frac{A_{v}+A_{v}^{c}}{D_{vv}+D_{vv}^{c}}\right)\\ & = & C\exp\left(-\int dv\;\frac{v\int_{\omega_{pe}/v}^{\infty }\frac{dk}{k}+\frac{m_{e}v^{3}}{T_{e}}\;\ln\Lambda \left(G\left(x_{e}\right)+\frac{T_{e}}{T_{i}}\;G\left(x_{i}\right)\right)}{\frac{4\pi^{2}}{m_{e}}\int_{\omega_{pe}/v}^{\infty }\frac{dk}{k}\;I_{L}\left(k\right)+v^{2}\;\ln\Lambda \left[G\left(x_{e}\right)+G\left(x_{i}\right)\right]}\right). \end{array}$$

This solution can be used to discuss either the thermal equilibrium that is attained by collisional process or the turbulent quasi-equilibrium state attained through collective fluctuations. If we ignore contributions from the collective fluctuations, that is, if we ignore the *k* integral terms in the numerator and denominator, then we have
(47)$$\begin{array}{lll} f_{e} & = & C\exp\left(-\frac{m_{e}}{T_{e}}\int dvv\;\frac{G\left(x_{e}\right)+\frac{T_{e}}{T_{i}}\;G\left(x_{i}\right)}{G\left(x_{e}\right)+G\left(x_{i}\right)}\right)\\ & = & C\exp\left(-\frac{m_{e}v^{2}}{2T}\right), \end{array}$$
where in going from the first to second equality, we have assumed $T_{e}=T_{i}=T$. This is the thermal equilibrium distribution, as expected. This regime where collective effects can be ignored, as compared to collisional effects, is supposed to be applicable on a time scale that is far longer than the time scale it takes for the plasma to enter the state of turbulent quasi-equilibrium. That is, the above approximation is supposed to be valid for temporal regime that is far longer than the stage of kappa distribution formation. On the other hand, if we ignore the collisional part dictated by lnΛ, then we have
(48)$$f_{e}=C\exp\left(-\frac{m_{e}}{4\pi^{2}}\int dvv\;\frac{\int_{\omega_{pe}/v}^{\infty }\frac{dk}{k}}{\int_{\omega_{pe}/v}^{\infty }\frac{dk}{k}\;I_{L}\left(k\right)}\right),$$
which is the same as (Equation 28). As we have already seen, this formal solution leads to the kappa distribution, provided the fluctuation spectrum is specified by the mathematical form given in (Equation 41). To sum up, the idea presented herein is to make use of combined collisional versus collective terms in the Fokker–Planck equation in order to discuss the transition from a kappa to a thermal distribution, but since we made the assumption of stationarity, ∂/∂t=0, the actual time scale for such a transition cannot yet be discussed in quantitative terms. Such a calculation requires actual numerical solutions to the equation.

In a recent work, Shizgal [31] made use of the general solution of the form (Equation 46) in order to discuss the formation of kappa and other non-Maxwellian distributions in the presence of an additional diffusion coefficient $D_{vv}$ provided by wave–particle interaction, in addition to the collisional diffusion coefficient $D_{vv}^{c}$. This is evident in Equation (Equation 7) of [31]. Note, however, that (Equation 46) is slightly more general in that the velocity friction coefficient is also made of two parts, the wave–particle-related term $A_{v}$ and the collision-related term $A_{v}^{c}$. Nevertheless, the essential conclusion from [31] is consistent with the present finding that the wave–particle interaction term within the generalized Fokker–Planck equation leads to kappa and non-Maxwellian distribution functions.

## 4. Summary

To summarize the essential findings of the present paper, we have argued for an inter-relationship that may exist between the non-extensive statistical description of plasma, in which long-range electromagnetic force is involved, and the quasi-steady-state turbulent plasma. Both descriptions share a common feature in that the equilibrium distribution function corresponds to the kappa distribution, or equivalently, the *q*-exponential distribution. In the non-extensive statistical approach, the *q* parameter is undetermined, but the plasma turbulence theory can be invoked in order to determine its value via the relationship q=(κ−1)/κ or q=κ/(κ+1) depending on whether one identifies $f_{q}$ [see (Equation 15)] as $f_{\kappa }∼{\left[1+v^{2}/\left(\kappa v_{T}^{2}\right)\right]}^{-\kappa }$ or $f_{\kappa }∼{\left\{1+v^{2}/\left[\left(\kappa +1\right)v_{T}^{2}\right]\right\}}^{-\kappa +1}$. If we adopt κ=9/4=2.25, then we find that q=5/9≈0.555⋯ or q=9/13≈0.6923⋯. We have also verified the theoretical prediction of κ=9/4=2.25 against spacecraft observations and found reasonable agreement. Finally, we have also briefly addressed the issue of including the effects of collisional relaxation in the general formalism.

## 5. Discussion

Before we close, we make a note of the fact that while the kappa or *q* distribution satisfactorily describes the measured electron distribution in solar wind where the core part of the velocity distribution function is adequately modeled by the Maxwell–Boltzmann (or Gaussian) profile while the energetic tail distribution is modeled by a quasi-inverse-power-law distribution [10,11,12,13], in many other regions of the space environment, often it is the low-energy spectrum of the particle distribution that exhibits quasi-inverse-power-law behavior while the high-energy portion of the particle population features an exponential cutoff. For instance, in the paper by Vasko et al. [33], the electron energy distribution measured in the inner magnetosphere region of the Earth by Van Allen Probes (VAP) spacecraft shows a distinctive exponential fall-off behavior for electron energy that exceeds 10 keV or so. Likewise, the proton distribution function measured by Advanced Composition Explorer (ACE) spacecraft features a similar inverse power law modified by an exponential cutoff for high velocity regimes [34].

Such a feature requires the modification of the standard kappa distribution. Recently, Yoon et al. [35] generalized the steady-state solution discussed earlier [18] to a situation where the high-velocity regime of the standard kappa distribution is modified by an exponential fall-off factor,

(49)$$f_{\kappa }^{\mathrm{modified}}\propto {\left(1+\frac{v^{2}}{\kappa v_{T}^{2}}\right)}^{-\kappa -1}\exp\left(-\frac{\alpha^{2}v^{2}}{v_{T}^{2}}\right).$$

Their work did not aim to describe observations such as those by Vasko et al. [33] or Fisk and Gloeckler [34], but rather, the work was prompted by an earlier theory by Scherer et al. [36], who sought to regularize the divergent behavior associated with the standard kappa distribution. It is well known that the standard kappa distribution has an eventual divergent velocity moment $\int dvv^{2n}f_{\kappa }$ for a certain value of *n* or higher, regardless of the value of the κ index. Scherer et al. [36] succeeded in “regularizing” the standard kappa model by attaching the exponential cutoff function, which led to all velocity moments being well defined. Yoon et al. [35] showed that the regularized kappa model of Scherer et al. [36] is also a legitimate steady-state solution of the Langmuir turbulence system. They were not concerned with explaining observed features such as those of refs. [33] or [34]. However, in view of the fact that the regularized (or modified) kappa model (Equation 49) has precisely the observed feature, such a model may be employed to explain the observations at least as a phenomenological fit.

Another issue regarding the present paper is that the theory is based on weak plasma turbulence. In plasma physics, the definition of weak turbulence involves the wave energy being sufficiently lower than the particle thermal energy. Space plasma is mostly in the state of weak turbulence since the measured electromagnetic wave energy is indeed much lower than the thermal energy, so it is appropriate to employ the assumption of weak turbulence. However, in laboratory plasmas or even in space plasmas when highly energetic events such as collisionless shocks are involved, the assumption of weak turbulence may be violated. To study the formation of kappa (or *q*-Gaussian) distributions for strong plasma turbulence is quite difficult, and with the present state of knowledge, the only viable means is to employ direct numerical computer simulations. Ryu et al. [37] carried out such a simulation in a regime of strong turbulence and demonstrated the formation of a kappa-like non-Maxwellian tail population, starting from the initial Gaussian plus beam distribution. However, their result is based upon a simulation and no theoretical interpretation is available yet.

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
