# Peer review of "Thermodynamic, Non-Extensive, or Turbulent Quasi-Equilibrium for the Space Plasma Environment"

_entropy, 2019, doi:10.3390/e21090820_

Round 1

Reviewer 1 Report

Review of Ms “Thermodynamic, non-extensive, or turbulent quasi equilibrium for space plasma environment” by P. H. Yoon In this work the Author provide a direct theoretical derivation of the q-parameter of Tsallis’ entropy in the case of long-range collective electromagnetic interaction. The work moves from micro-physics considerations and derives a number for the q-parameter which is very similar to the one that it is observed in the case of real turbulent space plasma environments. The paper is surely interesting but there is a major issue that could prevent the publication of this work. Indeed, I am not convinced that this paper presents explicitly new results in respect to the previous paper by the same Author, appeared in JGR in the 2004 (ref. 19). There a similar derivation of the k-distribution for electrostatic turbulence is proposed and, as in this work, the number for the exponent of the k-distribution is found to be the same, i.e., k = 2.25. Furthermore, the Author does not discuss studies on similar relationships between k-distribution and Tsallis’ statistics, as, for instance that by B. D. Shizgal, Phys. Rev. E, 97, 052144 (2018). According to this general argument, I believe that the Author should clearly clarify these majors point before the paper could be considered for publication in Entropy. Some other questions 1) One question concern the correct definition of additivity and extensivity. These two concepts, although they are generally confused, have a different meaning. Additivity refers to the the sum of two subsystems as correctly reported in Eq. (3). Extensivity refers to the dependence of the entropy on the number N of particles, thus I believe that the Authors should take care on the use of this terminology in the introduction so to avoid a confusion. Please, check the text and make the necessary corrections. 2) Why on Eq. 13 the Author defines k = 1/(1-q) instead of k = q/(1-q) so that the Vasyliunas’ kappa distribution is equivalent to the definition of Eq. (12) ? 3) Another issue in the Introduction and in other parts of the text deals with the usage of the word “quasi-equilibrium”. What is the meaning of this word ? Do we mean that we are in the limit of validity of linear non-equilibrium thermodynamics, i.e., in proximity of equilibrium where the Onsager relations are valid ? or it means “stationary non-equilibrium state”, such as the states corresponding to a kinetic thermodynamic branch which is not directly related to the equilibrium one ? Please clarify the usage of this term. 4) In Section 2 it could be immediately state that Eq. 14 is a Fokker-Planck equation instead of a plasma kinetic equation. Indeed, this Eq. does not have the form of Vlasov’s or Boltzmann’s equation but conversely as the corresponding Fokker-Planck equation in absence of external forces and/or spatial gradients. I suggests to spend some more word to clarify this point for non-specialist readers.

Author Response

I am uploading my response by pdf file.

Reviewer 2 Report

The manuscript is an interesting demonstration that power-law distribution functions often observed in the space plasma can arise in a process of relaxation of a turbulent plasma. The Author demonstrates that by adjusting the spectrum of fluctuations, we can find that stationary solutions of the quasi-linear/nonlinear equations are power-law distributions. The Author also provides a brief discussion of how these results are related to the space plasma observations. I am positive to recommend the publication of the manuscript, though I have several comments listed below.

Minor comments:

1. line 56-57: there is a recent paper by Tong et al., ApJL, 2019 (doi: https://doi.org/10.3847/2041-8213/aaf734) that clearly demonstrates the point mentioned by the Author and presents power-law indexes and other parameters of the electron velocity distribution functions in the solar wind.

2. The Author mentions that the velocity distribution functions in the space plasma are often power-law, especially at suprathermal energies. We would like to draw the Authors attention that “inverse” situation is also observed in the space plasma. For example, in the Earth’s inner magnetosphere we often observe f(E)~E^alpha*exp(-E/T), where alpha is between -1 and -2: lower-energy electrons have power-law distribution and higher-energy electrons are Maxwellian (see Vasko et al., Geophys. Res. Lett., 2017, doi: https://doi.org/10.1002/2017GL074026). Could these distributions be also produced by some turbulence? It seems that by adjusting the spectrum I(k), the Author can generate any stationary distribution function, right? It would be nice to discuss that not only power-law (i.e. kappa), but also other velocity distributions functions are observed in the space plasma.

editorial comments:

1. line 5: remove “but”

2. line 30: “sufficient”à “is sufficient”

Author Response

Please see the attached pdf file.

Round 2

Reviewer 1 Report

Referee's Report on Ms. entropy-526842, Thermodynamic, non-extensive, or turbulent quasi equilibrium for space plasma environment, by P. Yoon.

This revised version of the paper "Thermodynamic, non-extensive, or turbulent quasi equilibrium for space plasma environment", has been greatly improved in respect to the previous one. The Author has considered all my comments and questions, and improved the new version considering all those points. I believe that the paper is now ready for publication in Entropy. Thus, my suggestion is publish in the present form.

Author Response

We thank the reviewer for the positive assessment.